# Low Doses of Ketamine and Melatonin in Combination Produce Additive Antidepressant-like Effects in Mice

**DOI:** 10.3390/ijms22179225

**Published:** 2021-08-26

**Authors:** Rosa Estrada-Reyes, Daniel B. Quero-Chávez, Citlali Trueta, Armida Miranda, Marcela Valdés-Tovar, Salvador Alarcón-Elizalde, Julián Oikawa-Sala, Jesús Argueta, Luis A. Constantino-Jonapa, Jesús Muñoz-Estrada, Margarita L. Dubocovich, Gloria Benítez-King

**Affiliations:** 1Laboratorio de Fitofarmacología, Dirección de Investigaciones en Neurociencias, Instituto Nacional de Psiquiatría Ramón de la Fuente Muñiz, Calzada México-Xochimilco 101, San Lorenzo Huipulco, Tlalpan, Ciudad de México 14370, Mexico; restrada@imp.edu.mx; 2Laboratorio de Neurofarmacología, Subdirección de Investigaciones Clínicas, Instituto Nacional de Psiquiatría Ramón de la Fuente Muñiz, Calzada México-Xochimilco 101, San Lorenzo Huipulco, Tlalpan, Ciudad de México 14370, Mexico; quero.rude.daniel@gmail.com (D.B.Q.-C.); armida.miranda.riestra@gmail.com (A.M.); mvaldes@imp.edu.mx (M.V.-T.); salvadorcito1@hotmail.com (S.A.-E.); oikawasala@gmail.com (J.O.-S.); jadclear@yahoo.com (J.A.); biologia0712@gmail.com (L.A.C.-J.); 3Departamento de Neurofisiología, Dirección de Investigaciones en Neurociencias, Instituto Nacional de Psiquiatría Ramón de la Fuente Muñiz, Calzada México-Xochimilco 101, San Lorenzo Huipulco, Tlalpan, Ciudad de México 14370, Mexico; ctrueta@imp.edu.mx; 4Department of Pharmacology and Cancer Biology, Duke University School of Medicine, Durham, NC 27710, USA; munozestradajesus@gmail.com; 5Department of Pharmacology and Toxicology, Jacobs School of Medicine and Biomedical Sciences, University at Buffalo (SUNY), 955 Main Street, Buffalo, NY 14203, USA; mdubo@buffalo.edu

**Keywords:** melatonin, ketamine, depression, antidepressant

## Abstract

Major depressive disorder is a disabling disease with the number of affected individuals increasing each year. Current antidepressant treatments take between three to six weeks to be effective with forty percent of patients being resistant to treatment, making it necessary to search for new antidepressant treatments. Ketamine, a phencyclidine hydrochloride derivative, given intravenously, induces a rapid antidepressant effect in humans. In mice, it causes increased neurogenesis and antidepressant-like effects. However, it also produces psychomimetic effects in humans and in rodents increases the locomotor activity. In contrast, melatonin, a hormone secreted by the pineal gland and synthesized in extrapineal sites, increases new neuron formation and causes antidepressant-like effects in adult rodents with no collateral effects. Here, we assessed the effects of a non-effective dose of ketamine in combination with melatonin (KET/MEL), both on neurogenesis as well as on the antidepressant-like effect in mice. Our results showed that KET/MEL combination increased neurogenesis and produced antidepressant-like effects without altering locomotor activity after both single and triple administration protocols. Our data strongly suggest that KET/MEL combination could be used to simultaneously promote neurogenesis, reverting neuronal atrophy and inducing antidepressant-like effects.

## 1. Introduction

Major depressive disorder (MDD) is a recurrent and disabling psychiatric illness, projected to be the leading cause of disability worldwide in 2020. Decreased neurogenesis has been associated with the diminished hippocampal volume found in subjects with MDD [1,2]. Neurogenesis and dendritogenesis are stimulated by selective serotonin-reuptake inhibitors (SSRI), which are currently used to treat major depression [3].

Melatonin (MEL) (N-acetyl-5-methoxy tryptamine), an indolamine, synthesized by the pineal gland during the dark phase of the photoperiod and in peripheral and central extrapineal sites [4], has neuroprotective effects as a potent free radical scavenger [5]. It also increases the expression of antioxidant enzymes and has anti-apoptotic actions [6,7,8]. The hippocampus, a region specialized in memory and cognition [9], is one of the main targets of MEL in the brain. In rodents, MEL administration is followed by augmented cognition, new neuron formation, and dendritogenesis in the dentate gyrus, where MT1 and MT2 receptors and stimulation of CaMKII have been implicated in these processes [10,11,12,13].

Antidepressant-like effects of MEL have been demonstrated in rodents subjected to chronic unpredictable mild stress. MEL at 10 mg/kg decreased the immobility time in the tail suspension test (TST) when compared to control mice [14]. Under severe stress, produced by the forced swimming test (FST), MEL at 4 and 16 mg/kg is also effective in decreasing the immobility time in mice, depending on the circadian time of its administration (zeitgeber times; ZT 0 = lights on; 12:12 L/D) [15]. Additionally, antidepressant-like effects produced by MEL were enhanced when this indolamine was administered 30 min before the animals start to increase plasma concentration of MEL (zeitgeber time; ZT = 11) and when MEL concentrations reach maximal levels (ZT = 18) [16].

The administration of a combination of MEL with fluoxetine, an SSRI currently used in the treatment of major depression, diminished the levels of free radicals generated by chronic administration of fluoxetine to rodents [17,18]. In addition, synergic antidepressant-like responses as well as increased neurogenesis and dendritogenesis in the hippocampus have been observed after a chronic administration for 14 days of the combination of MEL (2 mg/kg) with citalopram (8 mg/kg) [11,12,19].

Currently available antidepressants are associated with frequent relapses and low remission rates, with 40% of patients being resistant to treatment. Antidepressant medication requires on average, between 6 to 8 weeks of treatment before reaching antidepressant efficacy. The long time required to attain antidepressant efficacy is especially of concern for MDD patients with suicidal ideation [20,21].

Recently, ketamine (KET), a phencyclidine hydrochloride derivative (2-(O-chlorophenyl)-2-(methylamino) cyclohexanone), and non-competitive N-methyl-D-aspartate (NMDA) receptor antagonist, has been used at sub-anesthetic doses to treat resistant MDD [22,23]. In rodents, KET at 3 mg/kg reduced depressive-like behavior [24] and enhanced neurogenesis. Both effects have been associated with glutamatergic actions. In humans, intravenous infusion of 0.5 mg/kg over 40 min of KET 5 days a week produces an antidepressant response in 24h (fast responders) [22,25,26]. Slow responder patients with resistant MDD treated with six infusions of 0.5 mg/kg KET given three times a week showed later antidepressant responses and remission within 14 days [22]. However, despite the antidepressant effect, KET produces dissociative and psychotomimetic effects, which have limited its use in the treatment of MDD [27,28].

In this work, we hypothesized that sub-effective doses of both MEL and KET, administered in combination, will have antidepressant-like effects and will produce new neuron formation in mice after a single administration at ZT18 or after the administration of melatonin at ZT 18 and ZT 11 followed by the administration of KET/MEL combination. Our findings indicate that the combination of minimal doses of KET/MEL elicit antidepressant-like effects and hippocampal neurogenesis without alterations in ambulatory activity.

## 2. Results

### 2.1. The Combination of Ketamine and Melatonin at Sub-Effective Doses Induced Antidepressant-like Effects after a Single Administration

Antidepressant effects of the KET/MEL combination were explored in two predictive behavioral mice models, the FST and the TST. The FST is a stress model leading to behavioral “despair” learned in the face of the impossibility of escape, while the TST induces mild stress in mice and is more sensitive to antidepressant drugs [29,30,31]. In the VEH-treated mice submitted to the FST, the immobility meantime was 59.0 ± 4.4 s (Figure 1A). Similar results were observed in the TST (Figure 1B). Imipramine administered acutely at 25 mg/kg, used as a drug reference, reduced immobility time from 63.30 ± 4.29 sec to 11.32 ± 0.86 s (82.58% reduction).

A single administration with KET at 3, 10, 20, 30 mg/kg to mice significantly decreased the immobility time compared to the VEH-treated mice in the FST (Figure 1A). In addition, we have previously shown that a single administration of MEL (4 or 16 mg/kg i.p.) 30 min before the FST does not reduce the immobility time in this test [15]. To examine whether the KET/MEL combination would be more effective than either drug alone in reducing the immobility behavior, we administered sub-effective doses of KET (1.5 mg/kg) and MEL (4 or 16 mg/kg, i.p.) simultaneously, 30 min before the FST [15]. KET at 1.5 mg/kg did not reduce the immobility time in mice summited to the FST (Figure 1A). However, the combination of this dose of KET (1.5 mg/kg) with MEL (4 or 16 mg/kg) significantly reduced the immobility time as compared with the control group (F_(2,29)_ = 39.00, *p* ≤ 0.001), reaching the best effect at 1.5/16 mg/kg of KET/MEL (Figure 1C). In the TST, this combination was also effective in reducing immobility behavior in comparison to the control group (Figure 1D)**.**

### 2.2. The Combination of Ketamine and Melatonin at Sub-Effective Doses Induced Antidepressant-like Effects after Two Previous Administrations of Melatonin

The antidepressant-like effects of the KET/MEL combination were tested at ZT 18 in mice previously administered with MEL in the middle of the previous dark phase (ZT 18) and one hour before the beginning of the dark phase (ZT 11), which corresponds to a time before the rising of plasma circulating levels of MEL (see Figure 2). The KET/MEL combination at 1.5/4 or 1.5/16 mg/kg administered at ZT = 18 after two previous doses of MEL decreased the immobility time compared with a group of animals triple-administered with the VEH at the same circadian times, from 59.05 ± 4.40 s to 16.51 ± 2.10 s (27.95%) and 36.15 ± 3.45 (61.21%), respectively (F_(2,31)_ =39.00, *p* < 0.001). Importantly, the lowest dose of MEL (4 mg/kg) in combination with KET produced a greater effect in comparison with the control group (Figure 2A). As shown in Figure 2B, the effect of KET/MEL in this triple administration protocol was higher than that of a triple administration of FLX at 15 mg/kg, which reduced the immobility time from 64.99 ± 6.15 s to 28.13 ± 1.04 s (43.35%) (t_(14)_ = 105.97, *p* ≤ 0.001).

### 2.3. Sub-Effective Doses of Ketamine and Melatonin Combination Do Not Modify the Mice Ambulatory Activity

Next, we assessed the effect of an acute administration of KET/MEL combination on the ambulatory activity using the OFT, in independent groups of mice that were treated similarly to the groups subjected to either FST or TST tests. KET at doses between 10–30 mg/kg, i.p. significantly increased the ambulatory activity (F_(5,47)_ = 2.99, *p* ≤ 0.001) and the number of rearings (F_(5,47)_ = 9.79, *p* ≤ 0.001) measured during a 5 min trial (Table 1). In contrast, KET at 1.5 and 3 mg/kg (which are doses that by themselves are non-effective in the FST and the TST), did not alter the mice ambulatory activity (Table 1). Moreover, the KET/MEL combination at either dose (1.5/4 or 1.5/16 mg/kg) did not affect the locomotor activity or the number of rearings upon a single administration (Table 1), nor after a double administration of MEL (Table 2)**.**

### 2.4. Single Administration of Ketamine/Melatonin Combination Increases the Expression of Doublecortin and Ki67 in the Mice Hippocampus

Neurogenesis occurs after antidepressant administration [32]. Thus, we explored if the KET/MEL combination was able to increase this process after a single administration, using MEL at a dose that produces antidepressant-like effects in mice (4 mg/kg) in the FST [15] and KET at non-effective doses [33]. Brain slices including the hippocampal dentate gyrus were labeled with a specific anti-doublecortin antibody. As shown in Figure 3D, abundant cells were observed with a doublecortin label. Quantitative data are shown in Figure 3E and a significant difference was observed in the number of cells/mm^3^ in the KET/MEL-treated mice (F_(3,25)_ = 263.3, *p* < 0.0001). We confirmed that neurogenesis occurs in the hippocampus of mice treated with KET/MEL by Western blot assays for doublecortin and Ki67, which are markers of this process. As shown in Figure 3F and 3G, the expressions of doublecortin and Ki67 were increased in the hippocampus of KET/MEL-treated mice, while no changes were observed in the groups of mice separately treated with either KET or MEL (F_(3,12) _= 12.8, *p* = 0.0005 for doublecortin and F_(3,8) _= 13.35, *p* = 0.02 for Ki67).

### 2.5. Treatment of Ketamine/Melatonin after Two Administrations of Melatonin Increases the Expression of Doublecortin in the Hippocampus of Mice

KET/MEL administration at ZT 18 after two injections of MEL administered at ZT 18 and ZT 11 increases the number of cells stained with the anti-doublecortin antibody in the dentate gyrus of the hippocampus (Figure 4C,D). Similarly, MEL administered at the same ZT times increases the number of doublecortin-positive cells (Figure 4B,D) (F_(2,48) _= 12.66 *p* < 0.0001). The relative amount of this protein determined by Western blot was not significantly different in both groups of mice (Figure 4E) (F_(2,15)_ = 2.454 *p* = 0.1197). By contrast, only the mice triple administered with MEL and the third administration with KET/MEL showed an increased number of nuclei stained with the anti-Ki67antibody (Figure 5C,D) (F_(2,48)_ = 8.943; *p* = 0.005). Data indicate that triple administration of MEL at ZT 18, ZT 11 and ZT 18 increase neurogenesis in the dentate gyrus of mice.

## 3. Discussion

In this study, we demonstrated antidepressant-like effects of the KET/MEL combination (at doses that are sub-effective by themselves), either with a single administration at ZT 18 or after two previous administrations of MEL at ZT 18 and ZT 11. Antidepressant-like effects were demonstrated in two predictive behavioral mice models, models that induce mild stress (TST) or that induce high stress (FST), which are useful tools for the characterization of antidepressant effects and antidepressant drugs [31]. In addition, we demonstrated that these two administration schemes induced neurogenesis in the dentate gyrus of the hippocampus.

Sub-anesthetic doses of KET administered to rodents produce rapid antidepressant-like effects with alterations in ambulatory activity [34]. In humans, a single intravenous administration of KET (0.1 to 0.5 mg/kg) causes fast antidepressant actions, with psychotomimetic and dissociative effects [35]. Both effects are lost after 3–12 days due to the short half-life of KET [35]. Importantly, we found that the combination of KET/MEL at smaller doses, which by themselves are non-effective, caused the antidepressant response without alterations in the ambulatory behavior in both a single administration at ZT 18, administered at this circadian time after two MEL injections at ZT 18 and ZT 11. These results indicate that MEL facilitates the fast antidepressant-like effect of KET, at sub-effective doses, and without the adverse effects of KET giving an additive effect.

MEL and KET/MEL were administered at the middle of the dark phase, when the endogenous profile of the pineal secretion of MEL reaches its highest levels [36]. Thus, antidepressant responses to KET at non-effective doses occur in conditions where MEL is endogenously present. In this sense, plasma-circulating levels of MEL are decreased in depressed patients [37] and in rodents it has been shown that MEL increases the responses to the SSRIs fluoxetine and citalopram [19,38].

We did not find significant differences in the antidepressant-like effects of the combination when it was administered either in a single injection or after two previous administrations of MEL. Thus, data support that endogenously or exogenously administrated MEL means that sub-effective doses of KET can be effective in producing antidepressant-like effects with no side effects. Moreover, our data suggest that it is necessary to reestablish endogenous MEL levels decreased in MD patients to get an optimal response to antidepressants. 

Additionally, in our study we used fluoxetine as a positive control since this drug had been previously demonstrated effective in reducing the immobility time in mice after its triple administration in the FST. Our results showed that fluoxetine and melatonin diminished the immobility time in the FST. 

Antidepressant compounds currently used in the treatment of MD increase neurogenesis in the dentate gyrus of the hippocampus [3,19]. In this study, we showed that a single administration of KET/MEL or the administration of this combination after two MEL injections in mice significantly increases hippocampal neuronal proliferation in vivo gauged by Western blot and immunostaining analysis of Ki67 and doublecortin marker. Importantly, no significant changes in neuronal proliferation were found after a single administration of either KET or MEL in comparison with mice treated with the vehicle. Previously, it was shown that only a chronic administration of MEL increased both neuronal survival and net neurogenesis in the mice dentate gyrus [39]. Our results support these findings because we did not find an increased proliferation after an acute administration of MEL. The data shown here indicate that acute administration in mice of KET in combination with MEL is sufficient to increase neuronal proliferation in the dentate gyrus similarly to what has been found for MEL under a chronic administration experimental scheme. Moreover, our data indicate that the antidepressant-like effects observed in mice are in part associated with an increased neurogenesis in the hippocampus. Together, data concur with the notion that antidepressant-like effects are associated with increased neurogenesis in the hippocampus [3,32].

The mechanism by which the KET/MEL combination increases neurogenesis and exerts its additive antidepressant-like effects is not known. KET is an antagonist of the N-methyl D-aspartate receptor (NMDAR). It acts on GABAergic interneurons, decreasing their activity in the brain prefrontal cortex, thus causing disinhibition of glutamatergic neurons that then activate AMPA receptors in post-synaptic neurons, and consequently BDNF secretion [39,40]. On the other hand, MEL stimulates MT_1_/MT_2_ receptors and in the rat habenula it acts on the glutamatergic signaling system [41]. Besides, it has been reported that MEL increases BDNF levels [42]. In this sense, KET antidepressant effects have been considered to be mediated by glutamatergic signaling, while MEL receptors are involved in the antidepressant-like effects of MEL in mice [42]. Further research is necessary to explain the mechanisms activated by the KET/MEL combination as well as the relevance of ZT administration to elicit antidepressant-like effects and neurogenesis. However, the results obtained here indicate that the combination of small doses of KET, non-effective in producing either antidepressant or locomotor changes, with MEL, a neuroprotective compound that produces robust antidepressant effects, would allow an optimal and safe combination for the fast and long-lasting treatment of mood disorders.

## 4. Materials and Methods

### 4.1. Animals and Pharmacological Treatments

Male Swiss Webster mice (25–35 g) were obtained from the vivarium of the Instituto Nacional de Psiquiatría Ramón de la Fuente Muñiz. Mice were managed following the specifications of Mexican Official Norm (NOM-062-ZOO-1999) and laboratory animal care (NIH publication # 85-23, revised in 1985) approved by the Instituto Nacional de Psiquiatría Ramón de la Fuente Muñiz, project number NC19127.0.

Mice were housed in groups of 8 per cage on a 12h reverse light/dark cycle (ZT 0 = lights on; 12 h light/12 h dark). All experiments were done in the dark under red light (12 lux/m^2^). Drugs were administered intraperitoneally (i.p.) in a volume of 10.0 mL/kg body weight.

Two types of drug administration were performed: (1) Mice were treated with a single dose of the vehicle (VEH; 0.06% ethanol solution), imipramine (IMI at 25 mg/kg), MEL (4 or 16 mg/kg), KET (1.5, 3, or 10 mg/kg), or the KET/MEL combination (1.5/4 or 1.5/16 mg/kg) at the middle of the dark phase (ZT 18), and the behavioral tests were performed 30 min later, at ZT 18.5 (Figure 1, upper scheme). (2) Mice received MEL (16 mg/kg) or FLX at ZT 18 (middle of the dark phase) and at ZT 11 (one hour before the beginning of the next dark phase), and then the KET/MEL combination was administered at the next ZT 18. Behavioral tests were performed 30 min later, at ZT 18.5 (Figure 2, upper scheme) [15].

### 4.2. Forced Swimming Test

Mice were individually placed into glass cylinders (height: 21 cm, diameter; 14.5 cm) containing 15 cm of water at 23 ± 1 °C and forced to swim for a 15 min period (pre-test) at ZT 18 to stimulate the immobility behavior. Then, the drugs were administered as explained above, and a second (test) swimming session was performed 30 min after the last drug administration (see schemes in Figure 1 and 2) [31,43].

### 4.3. Tail Suspension Test (TST)

Mice were treated with a single administration of the KET/MEL combination as described above. They were held individually by the tail for 5 min and suspended 50 cm above the surface of a wooden box by adhesive tape placed 1 cm from the tip of the tail. Immobility behavior was scored for 5 min when the mouse remained passively hung and completely motionless [44].

### 4.4. Open Field Test (OFT)

The ambulatory activity of independent groups of mice treated as described above was measured in an opaque-plexiglass box (40 × 30 × 20 cm), which was divided into 12 equal squares (11 × 11 cm). The animal was placed in a corner of the cage and videotaped over a 5-min period. The number of times the mice entered each square (counts number) and the number of times they stood on their hind legs (rearing number) were registered as described in [45].

### 4.5. Immunohistochemistry and Tissue Processing

Adult mice received one or three intraperitoneal injections of either VEH, KET, MEL or KET/MEL as described in the treatment section before the FST. After the test, mice were anesthetized with sodium pentobarbital (42 mg/kg) and perfused with (30 mL) phosphate buffer saline followed by 30 mL of 4% paraformaldehyde solution. Brains were dissected and post-fixed in 4% paraformaldehyde during 24 h before being transferred to a 30% sucrose cryopreservation solution.

Brains were embedded in OCT (optimal cutting temperature; Thermo Scientific, Waltham, MA, USA) compound and sectioned in the coronal plane at a thickness of 30 µm with a sliding microtome (Microm HM525, Thermo Scientific, MA, USA). The sections were treated with antigen retrieval buffer (10 mM Na Citrate, 0.05% Tween 20 pH 6.0) for 1 h at 37 °C when the anti-doublecortin antibody was used and blocked with 5% heat-inactivated donkey serum. Slices were permeabilized with 0.05% triton-X100 BSA 1% in PBS when anti-doublecortin antibody was used or with 0.2% of triton-X100 for anti-Ki67 staining. Tissue sections were washed with PBS and stained following a free-floating immunostaining method with anti-doublecortin (1:500 Abcam, Ab207175, UK) antibody or anti-Ki67 antibody (1:500; Abcam, Ab15580, UK) and fluorescently conjugated secondary antibody DyLight 488 (1:500, Life Technologies, Thermo Scientific, MA, USA), followed by DAPI counterstaining (1.5 µg/mL). Then, tissue sections were mounted on slides with Vectashield (Vector Laboratories Inc., Burlingame, CA, USA). Immunofluorescent images were acquired with a NIKON ECLIPSE Ti microscope and 20X objective, using the same exposure times for all sections. The proportion of doublecortin-positive cells was analyzed throughout the rostro-caudal extent of the sub granular zone of the dentate gyrus. All images were analyzed using NIS-ELEMENTS software from NIKON.

### 4.6. Western Blot

Hippocampi were dissected from mice treated as described before and subjected to the FST. The tissue was homogenized in ice-cold RIPA buffer (50 mmol/L Tris-HCl, pH 7.4, 150 mmol/L NaCl, 0.1% NP-40, 1 mmol/L EDTA, 0.25% sodium deoxycholate, 1 mmol/L PMSF, 1 μg/mL, aprotinin, 1 μg/mL leupeptin, 1 μg/mL pepstatin, 1 mmol/L NaF, 1 mmol/L Na_3_VO_4_) for 60 s using a ultrasonicator (Cole Palmer, IL, USA) set at 60 Hz. Samples were maintained on ice and centrifuged at 12,000 rpm at 4 °C for 15 min. Cleared lysates were collected, and total protein was quantified with Lowry’s method [46]. Samples were diluted in 2× Laemmli SDS sample buffer and boiled for 5 min at 97 °C. Equal amounts of protein were separated by SDS-PAGE in 10% polyacrylamide gels (Stain-free BIO-RAD, Hercules, CA, USA) and transferred to low fluorescence PVDF membranes at 120 V, for 3 min for doublecortin (Trans-Blot Turbo, BIO-RAD, CA, USA). Membranes were blocked in 5% BSA in phosphate-buffered saline with 0.05% Tween 20 (PBS-T) for 30 min and then incubated with an anti-doublecortin (1:500; Abcam, Ab207175, UK) or anti-Ki67 antibody (1:500; Abcam15580, UK), followed by a (1:10,000 dilution) secondary antibody coupled to DyLight 680 (Invitrogen, SA5-10042, Thermo Scientific, MA, USA). Primary and secondary antibodies were incubated in PBS-T BSA 5% overnight at 4 °C and 1h at RT, respectively. After extensive washes with PBS-T, secondary antibodies were detected by ChemiDoc^TM^ MP Imaging System (BIO-RAD, CA, USA and the analysis of images was done using the Image Lab software version 5.2.1 build 11, 2014 (BIO-RAD Laboratory, CA, USA). Normalization of DCX and Ki67 relative number of proteins by Western blot was done by maximum and minimum values which were calculated with the smallest (min) value of the group and the largest (max) value of the group, applied to the formula: $\frac{x-\min}{\max-\min\;}$. The results are presented as scale 0–1. 

### 4.7. Statistical Analysis

Sample sizes of animals for behavioural tests were determined with the GRANMO program that contemplates the alpha risk of 0.05 and beta risk of 0.2 in a two-sided test, with the result being that 8 subjects were necessary in both the first and second group to find any statistically significant differences between groups. A proportion difference is expected to be 0.99 in group 1 and 0.4 in group 2. We anticipated a drop-out rate of 20%. The ARCSINUS approximation was that 20% of mice were not responders, and the response variability and typical standard deviation had a bilateral or two-tailed comparison. The number of samples for in vitro assays for the one-way ANOVA model was calculated with the R program package “OPDOE” (R Core Team (2021). R: A language and environment for statistical computing. Vienna, Austria). For the experiments we used an α=0.05, β=0.2, δ=2 for Ki67 experiments and a δ=2 for DCX experiments. Finally, the number of levels for the variable in the model were 4 (VEH, MEL, KET, KET/MEL) for Figure 3 and 3 levels (VVV, MMM, MMK/M) for Figure 4 and Figure 5.

Data that met the criteria of normality (Kolmogorov–Smirnov test), a Student´s t-test or a one-way analysis of variance (ANOVA) were done. Dunnett´s, Bonferroni´s or Tukey’s tests for multiple comparisons vs control group were applied when the ANOVA showed significant differenced; *p*-values ≤ 0.05 were statistically significant. Data and statistical analysis were carried out using the Sigma Plot Program (version 12.3, CA, USA). Graphics were made with GraphPad Prism 7 (CA, USA).

## 5. Conclusions

Our results show that the KET/MEL combination produces antidepressant-like effects in behavioral tests and increased neurogenesis in the dentate gyrus of the mouse hippocampus. The antidepressant-like effects exerted by the KET/MEL combination at low doses opens the possibility of a best approach for the treatment of depressive disorders, with the additional benefit of a rapid response onset within two hours of administration without adverse effects, in comparison to the weeks or months required for standard medication [46].

## 6. Patents

G.B.-K., R.E.-R., M.L.D., C.T., D.B.Q.-C. and M.V.-T. are co-inventors on a patent for the use of the ketamine and melatonin combination in the treatment of mood disorders and assigned their patent’s rights to the “Instituto Nacional de Psiquiatría Ramón de la Fuente Muñíz”, but will share a percentage of any royalties that may be received with the Institution.

## Figures and Tables

**Figure 1 ijms-22-09225-f001:**
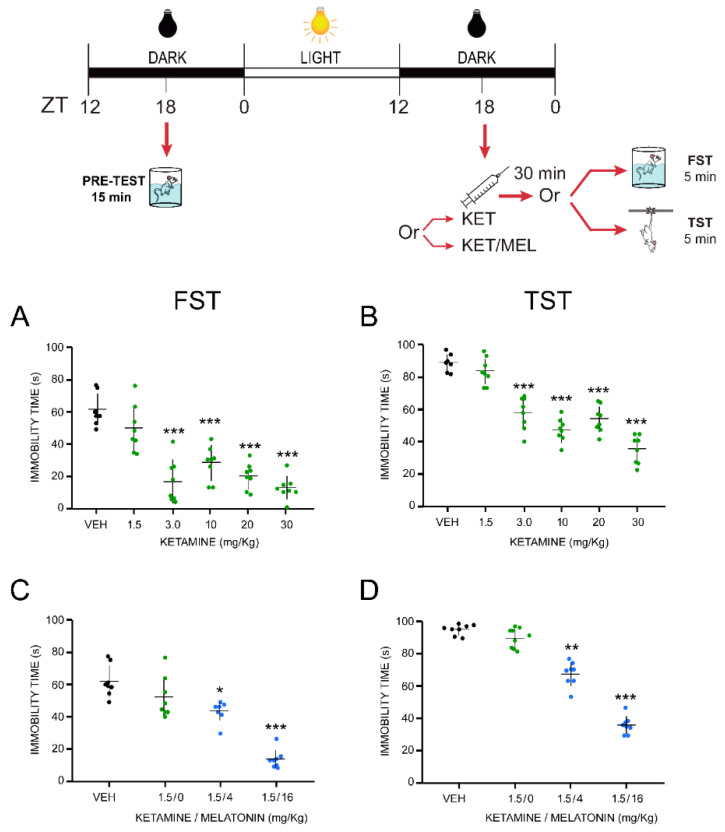
Effect of acute treatment with ketamine and melatonin and their combination on the forced swimming and tail suspension tests in mice. Animals (*n* = 8 per group) were injected with the VEH or a single dose of KET (**A**,**B**) (1.5 mg/kg, i.p.) or the combination of KET with either 4 or 16 mg/kg (i.p.) of MEL at ZT 18, 30 min before the start of either the forced swimming (**C**) or the tail suspension tests (**D**) (see Methods section). Data are expressed as the mean ± standard error of the mean (SEM) of independent groups (*n* = 8). Data were analyzed using a one-way ANOVA followed by Bonferroni’s post-test. * *p* ≤ 0.05, ** *p* ≤ 0.01, *** *p* ≤ 0.001 compared with vehicle (VEH) control group.

**Figure 2 ijms-22-09225-f002:**
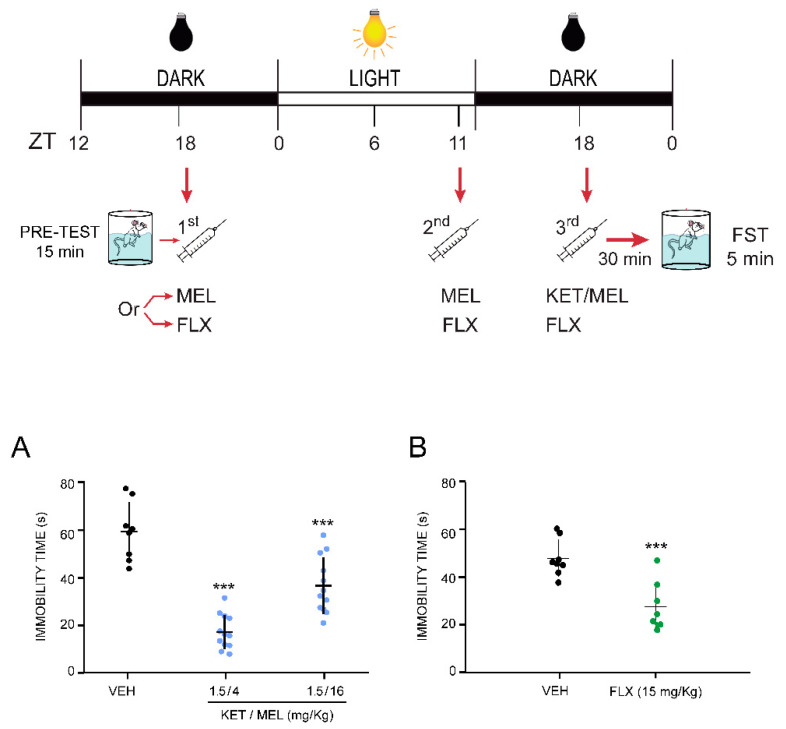
Effect of the co-administration of ketamine and melatonin after two administrations of melatonin to mice submitted to the forced swimming test. First, 24 h before starting the behavioral test, mice (*n* = 8 per group) were i.p. injected at ZT 18 with either 4 or 16 mg/kg of MEL (first administration). A second dose of MEL was i.p. administrated at ZT 11 of the light cycle. Finally, in a third injection, mice received MEL at either of 4 or 16 mg/kg in combination with a non-effective dose of KET (1.5 mg/kg) at ZT 18 (**A**). Another independent group was given 15 mg/kg (i.p.) fluoxetine as a reference drug, following the same administration protocol (**B**). The forced swimming test was applied 30 min after the last injection. Results are the mean + error standard of the mean (SEM) of independent determinations of 8 to 12 mice per group. Data were analyzed using a one-way ANOVA followed by the Bonferroni post-test. (**A**) when compared with the vehicle (VEH) control group. For (**B**) a Student’s t-test was done (t_(14)_ = 105.97, *p* ≤ 0.001). *** *p* < 0.001.

**Figure 3 ijms-22-09225-f003:**
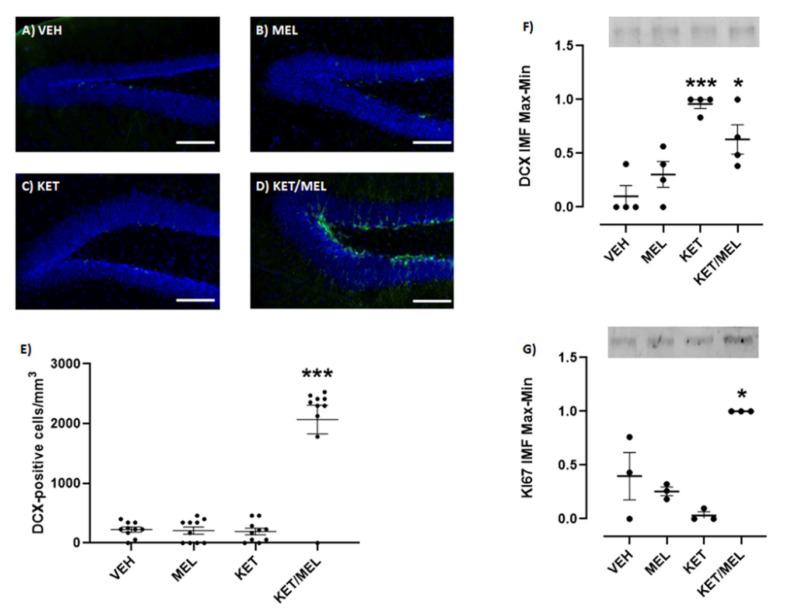
Effect of an acute administration of KET/MEL combination on the doublecortin and K167 expression in the mice hippocampus. Animals (*n* = 3 per group) were i.p injected with the VEH or a single dose of MEL (4 mg/kg), KET (1.5 mg/kg), or the combination of KET/MEL (1.5/4 mg/kg) at ZT 18, 30 min before the FST. Then, they were anesthetized and perfused as described in methods. The brains were sectioned in 30 µm slices and immunostained with an anti-doublecortin antibody and a secondary DyLight 488 antibody. Nuclei were stained with DAPI (**A**–**D**). The number of doublecortin-positive cells were counted in the subgranular zone in the dentate gyrus (**E**), each point representing a determination in one brain slice image. A total of 18 slice images (two images from two different hippocampal zones) derived from 9 brain slices of 3 mice per group were assessed. Data was analyzed using a one-way ANOVA, F_(3,25)_ = 263.3, *p* = 0.003. Another mice group (*n* = 4 for DCX and *n* = 3 for Ki67, per group) was similarly injected and after the FST they were sacrificed. The hippocampal region was dissected, homogenized, and analyzed by Western blot. Doublecortin and Ki67 bands were immunostained with specific antibodies and detected by secondary DyLight 680 antibodies. Quantitative graphs for doublecortin (DCX) and Ki-67 are shown in panels (**F**,**G**), respectively. Samples were run for duplicate. Each point represents two technical replicates. Data were analyzed using a one-way ANOVA followed by a Tukey post-test. Data were normalized by total protein as well as by maximal and minimal values. (F_(3,12)_ = 12.8, *p* = 0.0005 for DCX and F_(3,8)_ = 13.35, *p* = 0.02 for Ki67). * *p* < 0.05, *** *p* < 0.001. Scale bar = 100 µm).

**Figure 4 ijms-22-09225-f004:**
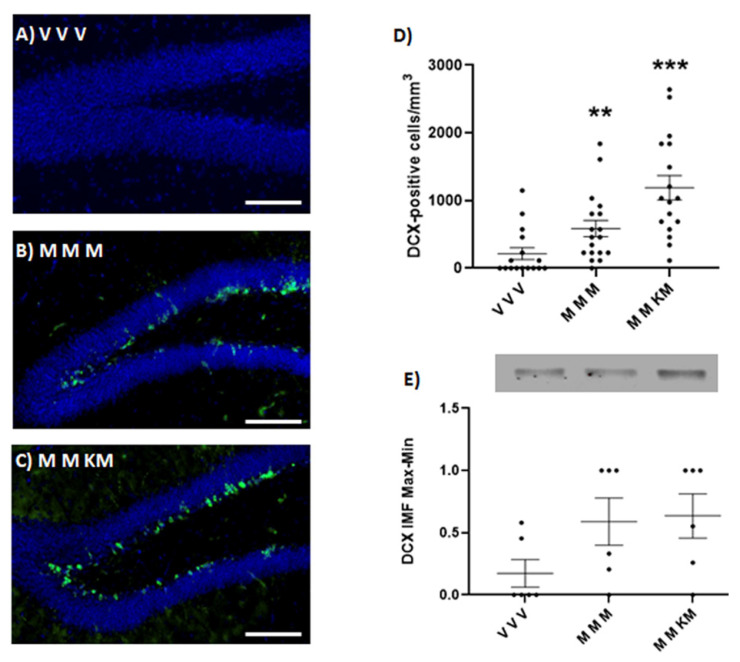
Effect of ketamine/melatonin combination after two administrations of melatonin on the expression of doublecortin in the mice hippocampus. Mice (*n* = 3 per group) were injected at ZT 18 with 4 mg/kg of melatonin (M) or the vehicle (V) as first administration. A second dose of melatonin (M) or the vehicle (V) was administered at ZT 11 of the light cycle. Finally, in a third injection mice received melatonin (4 mg/kg) in combination with a non-effective dose of KET (1.5 mg/kg) at ZT 18 (KM) or solely melatonin (M). Animals were anesthetized and perfused. The brains were sectioned in 30 µm slices and immunostained with an anti-doublecortin antibody and a secondary DyLight 488 antibody. Nuclei were stained with DAPI (**A**–**C**). The number of doublecortin-positive cells were counted in the sub granular zone in the dentate gyrus (**D**) (F_(2,48) _= 12.66; *p* < 0.0001). Each point represents a determination in one brain slice image. A total of 18 slice images (two images from two different hippocampal zones) derived from 9 brain slices of 3 mice per group were assessed. Another group of mice (*n* = 3) was similarly injected and after the FST they were sacrificed. The hippocampal region was dissected, homogenized, and analyzed by Western blot. Doublecortin was immunostained with a specific antibody and detected by secondary DyLight 680 antibody. The quantitative graph is shown in panel (**E**). Samples were run for duplicate. Each point represents one technical replicate derived from 3 animals per group. Data were analyzed using a one-way ANOVA followed by Tukey post-test. Data were normalized by total protein as well as by maximal and minimal values. (F_(2,15)_ = 2.454; *p* = 0.1197). ** *p* < 0.01, *** *p* < 0.001. Scale bar = 100 µm.

**Figure 5 ijms-22-09225-f005:**
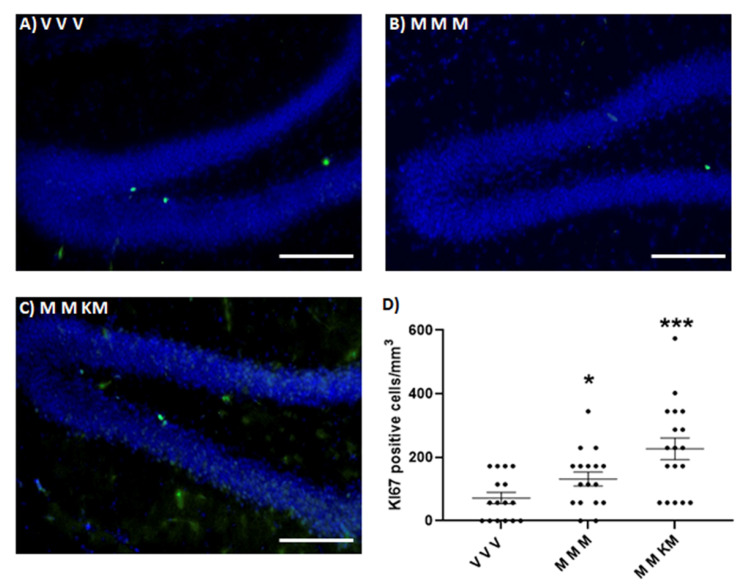
Effect of ketamine/melatonin combination after two administrations of melatonin on the expression of Ki67 in the mice hippocampus. Animals (*n* = 3 per group) were injected at ZT 18 with 4 mg/kg of melatonin (M) or the vehicle (V) as first administration. A second dose of melatonin (M) or the vehicle (V) was administered at ZT 11 in the light cycle. Finally, mice received a third injection of melatonin (4 mg/kg) in combination with a non-effective dose of KET (1.5 mg/kg) at ZT 18 (KM) or solely melatonin (M). Animals were anesthetized and perfused. The brains were sectioned in 30 µm slices and immunostained with an anti-Ki67 antibody and a secondary DyLight 488 antibody. Nuclei were stained with DAPI (**A**–**C**). The number of Ki67-positive cells were counted in the sub granular zone in the dentate gyrus (**D**). Each point represents a determination in one brain slice image. A total of 18 slice images (two images from two different hippocampal zones) derived from 9 brain slices of 3 mice per group were assessed. Data were analyzed using a One-Way ANOVA followed by Tukey post-test, (F_(2,48) _= 8.943; *p* = 0.0005) * *p* < 0.05, *** *p* < 0.001. Scale bar = 100 µm.

**Table 1 ijms-22-09225-t001:** Effects of melatonin (MEL), ketamine (KET) and MEL/KET combination acute treatments on the open field test.

Treatment (mg/Kg) ^1^	Counts Number	Rearings Number
Control	36.87 ± 3.76	30.75 ± 2.57
KET 1.5	40.12 ± 3.23	28.62 ± 3.15
KET 3	45.25 ± 4.29	28.25 ± 2.66
KET 10	56.62 ± 4.36 **	45.25 ± 2.85 **
KET 20	56.00 ± 3.62 **	45.50 ± 3.70 **
KET 30	60.12 ± 1.60 ***	45.37 ± 1.71 ***
	F_(5,47)_ = 7.260, *p* < 0.001	F_(5,47)_ = 9.797, *p* ≤ 0.001
Control	43.87 ± 4.42	34.50 ± 3.95
KET 1.5/MEL 4	40.11 ± 3.61	26.66 ± 2.21
KET 1.5/MEL 16	40.90 ± 3.60	27.30 ± 3.03
	F_(2,26)_ = 0.248, *p* = 0.783	F_(2,26)_ = 1.859, *p* = 0.178

^1^ Data are expressed as the mean ± standard error of the mean (SEM) of independent determinations with 8 to 10 mice per group. Differences between groups were analyzed using a one-way ANOVA, followed by Dunnett’s post-test. ** *p* ≤ 0.01, *** *p* ≤ 0.001 when compared with the vehicle control group.

**Table 2 ijms-22-09225-t002:** Effects of melatonin (MEL), ketamine (KET) and MEL/KET combination treatments in mice following the triple administration protocol on the open field test.

Treatment (mg/Kg) ^1^	Counts Number	Rearings Number
Control	41.12 ± 4.71	25.37 ± 2.18
MEL4	47.20 ± 5.20	29.11 ± 2.16
MEL 16	42.62 ± 5.97	26.37 ± 4.60
KET 1.5	41.40 ± 3.97	28.00 ± 2.53
	F_(3,35)_ = 0.279, *p* = 0.840	F_(5,33)_ = 0.865, *p* = 0.470
Control	43.37 ± 4.71	33.25 ± 2.72
KET 1.5/ MEL 4	36.87 ± 3.03	28.37 ± 3.25
KET 1.5 /MEL 16	37.00 ± 5.56	26.75 ± 3.22
	F_(2,23)_ = 0.664, *p* = 0.525	F_(2,23)_ = 1.210, *p* = 0.318

^1^ Data are expressed as the mean ± standard error of the mean (SEM) of independent determinations with 8 to 10 mice per group. Differences between groups were analyzed using a one-way ANOVA, followed by the Dunnett´s post-test.

## Data Availability

The data presented in this study are available on request from the corresponding author.

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
