# Peer review of "Low Doses of Ketamine and Melatonin in Combination Produce Additive Antidepressant-like Effects in Mice"

_ijms, 2021, doi:10.3390/ijms22179225_

Round 1
Reviewer 1 Report
The study is interesting and novel. However, the number of animals per group used is too small to draw the conclusions. Typically, n= 3 or 4 is too small to be analyzed by One-Way ANOVA and also to perform a test of normal distribution. How the authors select the sample size? Moreover, although the n is = 8 for example in figure 2, the number of “points” in the graph is different ( 11-12 for the groups KET/MEL). The same happens in figure 3 and 4. Classically, n represents the number of animals/group and does not mean number of replicates.
Other points are:
- The rationale for using fluoxetine in a sub-chronic protocol is not clear since fluoxetine requires repeated treatment to exert an antidepressant effect.
Moreover, in figure 2B, the vehicle group showed an immobility time quite different to that of vehicle group in figure 2A.
- In the table 2, given the control group is the same, authors should analyze all experimental groups together.
- It is not clear what “maximal and minimal values” means in the normalization methods. Stain-free membranes should be shown, too.
- It is not clear why MEL and KET/MEL were administered at the middle of the dark phase.
- FST and TST are behavioral tests, not models of depression.
- Discussion is too speculative since no data are shown on the mechanism of action of KET/MEL combination
Reviewer 2 Report
Submitted manuscript presents data originating from the concept that co-administration of melatonin and ketamine may exert fast antidepressant action in experimental animals. The behavioral part of the study is well planned, however, there are concerns regarding methodological approach. Therefore, the manuscript cannot be published in the present form.
Major comments:
1) The extent of neurogenesis was studied in brain specimens obtained right after the 2nd behavioral examination (forced swimming test) i.e. approx. 45 min. after the administration of third dose of treatment. However, the description of Figure 3 states that: “Animals (n=3 per group) were i.p injected with the VEH or a single dose of MEL (4 mg/kg), KET (1.5 mg/kg), or the combination of KET/MEL (1.5/4 mg/kg) at ZT 18, 30 min before the FST. Then…” and „Another mice (n=4 per group), were similarly injected and after the FST they were sacrificed. The hippocampal region was dissected, homogenized, and analyzed by Western blot.” Thus, the authors did not run immunochemistry and Western blots in parallel, on samples from the same animal, which is easily attainable (one hemisphere is used for each procedure), but on different animals. The group of animals used for Western blot was subjected to forced swimming test in contrast to the group designated for the immunochemistry what could affect the outcome.
2) The number of animals used for studying The effect of an acute administration of KET/MEL combination on the double-cortin and K167 expression in the mice hippocampus (Figure 3) is N=3. However, on Figure 3, panel E, the authors show 9 data points – where do they come from? If the data come from different slices but from the same animal, it seems like a manipulation. Such methodology could easily lead to a situation in which the authors would study 20 slices from one animal and subsequently show 20 data points, run the statistics on N=20 and conclude that data are scientifically valid. Similar manipulation has been used in Figure 4D and 5D.
Author Response
Answers to Reviewer 2.
Q1. The authors did not run immunochemistry and Western blots in parallel, on samples from the same animal.
We agree this observation. Run samples in parallel was not possible because we used intracardiac perfusion with the fixative paraformaldehyde for processing samples for immunohistochemistry. This procedure allows the obtaining of well-preserved brain tissue for fluorescence microscopy detection of proteins of interest. Once the tissue is fixed with paraformaldehyde, it is not possible to obtain homogenates of the contralateral brain hemisphere for Western blot processing. However, animal groups for these experiments were treated with the same pharmacological scheme and submitted the FST test before collecting tissue samples for both Western blot and fluorescent-base immunochemistry analysis.
Q2. The group of animals used for Western blot was subjected to forced swimming test, in contrast to the group designated for immunochemistry what could affect outcome.
All the animals were submitted to the FST before the obtaining of tissue samples for Western blot or immunochemistry analysis.
Q3: The number of animals used for the s study for immunohistochemistry is N=3. However, the authors show 9 data points-where do they come? Similar manipulation has been used in Figure 4D and 5D.
Each point in the graphs of figures 3E, 4D, and 5D represents a determination in one brain slice image. A total of 18 slice images (two images from two different hippocampal zones) derived from 9 brain slices of 3 mice per group were assessed. For clarity purposes, these lines were added in figure legends 3, 4, and 5. Missing points in the figures are due to technical problems, so at least 9 points per treatment were analyzed.
The number of samples for in vitro assays for the one-way ANOVA model was calculated with the R program package “OPDOE”. For the experiments we used an , , for Ki67 experiments and a for DCX experiments. Finally, the number of levels for the variable in the model were 4 (VEH, MEL, KET, KET/MEL) for figure 3 and 3 levels (VVV, MMM, MMK/M) for figures 4 and 5. This paragraph was added at the end of the 4.7 section, at line 408.
The sample size for the experiments in figure 3 was n=4 and for experiments in figures 4 and 5 were n=5. However, when significance was achieved (p<0.05) experiments were stopped, that’s why we have almost all experiments with an n=3.
Round 2
Reviewer 1 Report
The authors have answered to all my concerns